# Missing Data Infill with Automunge

## Abstract

Missing data is a fundamental obstacle in the practice of data science. This paper surveys a few conventions for imputation as available in the Automunge open source python library platform for tabular data preprocessing, including "ML infill" in which auto ML models are trained for target features from partitioned extracts of a training set. A series of validation experiments were performed to benchmark imputation scenarios towards downstream model performance, in which it was found for the given benchmark sets that in many cases ML infill outperformed for both numeric and categoric target features, and was otherwise at minimum within noise distributions of the other imputation scenarios. Evidence also suggested supplementing ML infill with the addition of support columns with boolean integer markers signaling presence of infill was usually beneficial to downstream model performance. We consider these results sufficient to recommend defaulting to ML infill for tabular learning, and further recommend supplementing imputations with support columns signaling presence of infill, each as can be prepared with push-button operation in the Automunge library. Our contributions include an auto ML derived missing data imputation library for tabular learning in the python ecosystem, fully integrated into a preprocessing platform with an extensive library of feature transformations, with a novel production friendly implementation that bases imputation models on a designated train set for consistent basis towards additional data.

## 1   Introduction

Missing data is a fundamental obstacle for data science practitioners. Missing data refers to feature sets in which a portion of entries do not have samples recorded, which may interfere with model training and/or inference. In some cases, the missing entries may be randomly distributed within the samples of a feature set, a scenario known as missing at random. In other cases, certain segments of a feature set's distribution may have a higher prevalence of missing data than other portions, a scenario known as missing not at random. In some cases, the presence of missing data may even correlate with label set properties, resulting in a kind of data leakage for a supervised training operation.

In a tabular data set (that is a data set aggregated as a 2D matrix of feature set columns and collected sample rows), missing data may be represented by a few conventions. A common one is for missing entries to be received as a NaN value, which is a special numeric data type representing "not a number". Some dataframe libraries may have other special data types for this purpose. In another configuration, missing data may be represented by some particular value (like a string configuration) associated with a feature set.

When a tabular data set with missing values present is intended to serve as a target for supervised training, machine learning (ML) libraries may require as a prerequisite some kind of imputation to ensure the set has all valid entries, which for most libraries means all numeric entries (although there are some libraries that accept designated categoric feature sets in their string representations).

Conventions for imputation may follow a variety of options to target numeric or categoric feature sets [Table 1], many of which apply a uniform infill value, which may either be arbitrary or derived as a function of other entries in the feature set.

Table 1: Imputation Conventions

| Imputation Value | Numeric | Categoric |
|---|:---:|:---:|
| mean | ✓ | |
| median | ✓ | |
| mode | ✓ | ✓ |
| adjacent cell | ✓ | ✓ |
| arbitrary (e.g. 0 or 1) | ✓ | ✓ |
| distinct activation | | ✓ |
| ML infill | ✓ | ✓ |

Other, more sophisticated conventions for infill may derive an imputation value as a function of corresponding samples of the other features. For example, one of many learning algorithms (like random forest, gradient boosting, neural networks, etc.) may be trained for a target feature where the populated entries in that feature are treated as labels and surrounding features sub-aggregated as features for the imputation model, and where the model may serve as either a classification or regression operation based on properties of the target feature.

This paper is to document a series of validation experiments that were performed to compare downstream model performance as a result of a few of these different infill conventions. We crafted a contrived set of scenarios representing paradigms like missing at random or missing not at random as injected in either a numeric or categoric target feature selected for influence toward downstream model performance. Along the way we will offer a brief introduction to the Automunge library for tabular data preprocessing, particularly those aspects of the library associated with missing data infill. The results of these experiments summarized below may serve as a validation of defaulting to ML infill for tabular learning even when faced with different types of missing data, and further defaulting to supplementing imputations with support columns signaling presence of infill.

Our contributions include an auto ML derived missing data imputation library for tabular learning in the python ecosystem, fully integrated into a preprocessing platform with an extensive library of feature transformations, extending the ML imputation capabilities of R libraries like MissForest [1] to a more production friendly implementation that bases imputation models on a designated train set for consistent basis towards additional data.

## 2  Automunge

Automunge [2], put simply, is a python library platform for preparing tabular data for machine learning, built on top of the Pandas dataframe library [3] and open sourced under a GNU GPL v3.0 license. The interface is channeled through two master functions: automunge(.) for the initial preparation of training data, and postmunge(.) for subsequent efficient preparation of additional "test" data on the train set basis. In addition to returning transformed data, the automunge(.) function also populates and returns a compact dictionary recording all of the steps and parameters of transformations and imputations, which dictionary may then serve as a key for consistently preparing additional data in the postmunge(.) function on the train set basis.

Under automation the automunge(.) function performs an evaluation of feature set properties to derive appropriate simple feature engineering transformations that may serve to normalize numeric sets and binarize (or hash) categoric sets. A user may also apply custom transformations, or even custom sets of transformations, assigned to distinct columns. Such transformations may be sourced from an extensive internal library, or even may be custom defined. The resulting transformed data log the applied stages of derivations by way of suffix appenders on the returned column headers.

Missing data imputation is handled automatically in the library, where each transformation applied includes a default imputation convention to serve as a precursor to imputation model training, one that may also be overridden for use of other conventions by assignment.

Included in the library of infill options is an auto ML solution we refer to as ML infill, in which a distinct model is trained for each target feature and saved in the returned dictionary for a consistent imputation basis of subsequent data in the postmunge(.) function. The model architecture defaults to random forest [4] by Scikit-Learn [5], and other auto ML library options are also supported.

The ML infill implementation works by first collecting a 'NArw' support column for each received feature set containing boolean integer markers (1's and 0's) with activations corresponding to entries with missing or improperly formatted data. The types of data to be considered improperly formatted are tailored to the root transformation category to be applied to the column, where for example for a numeric transform non-numeric entries may be subject to infill, or for a categoric transform invalid entries may just be special data types like NaN or None. Other transforms may have other configurations, for example a power law transform may only accept positive numeric entries, or an integer transform may only accept integer entries.

This NArw support column can then be used to perform a target feature specific partitioning of the training data for use to train a ML infill model [Fig 1]. The partitioning segregates rows between those corresponding to missing data in the target feature verses those rows with valid entries, with the target feature valid entries to serve as labels for a supervised training and the other corresponding features' samples to serve as training data. Feature samples corresponding to the target feature missing data are grouped for an inference operation. Note that for cases where a transformation set has prepared a target input feature in multiple configurations, those derivations other than the target feature are omitted from the partitions to avoid data leakage. A similar partitioning is performed for test data sets for ML infill imputation, although in this case only the rows corresponding to entries of missing data in the target feature are utilized for inference. As a further variation available for any of the imputation methods, the NArw support columns may themselves be appended to the returned data sets as a signal to training of entries that were subject to infill.

## ML infill

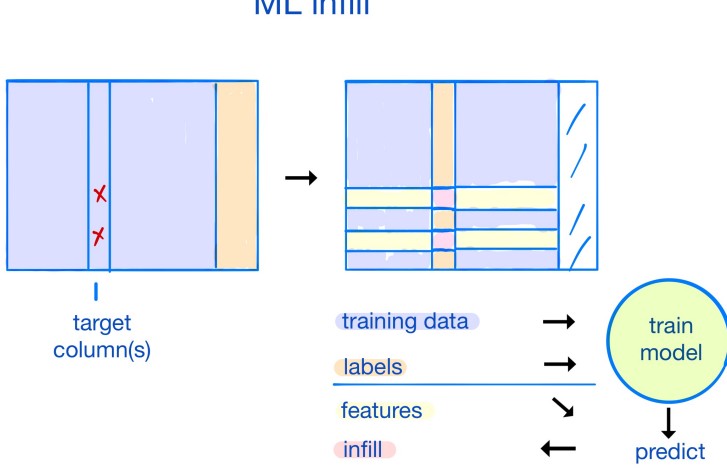

Figure 1: ML Infill partitioning

There is a categorization associated with each preprocessing transformation category to determine the type of ML infill training operation, for example a target feature set derived from a transform that returns a numeric form may be a target for a regression operation or a target feature set derived from a transform that returns an ordinal encoding may be a target for a classification operation. In some cases a target feature may be composed of a set of more than one column, like in the case of a set returned from a one-hot encoding. For cases where a learner library does not accept some particular form of encoding as valid labels there is a conversion of the target feature set for training and an inverse conversion after any inference, for example it may be necessary to convert a binarized target feature set to one-hot encoding or ordinal encoding for use as labels in different auto ML frameworks.

As may be particularly beneficial in cases with high prevalence of missing data across features, the sequential training of feature imputation models may be iterated through repeated rounds of imputations. For instance in the first round of model trainings and imputations the models' performance

may be slightly degraded by high prevalence of missing data populated with the initial transformation function imputation conventions in surrounding features, but after that first round of imputations a second iteration of model trainings may have slight improvement of performance due to the presence of ML infill imputations, and similarly ML infill may benefit from any additional iterations of model trainings and imputations. In each iteration the sequence of imputations between columns are applied in an order from features with highest prevalence of missing data to least. The library defaults to a single round of imputations, with the option to specify an additional iteration quantity.

The final trained models for each target feature, as derived from properties of a designated train set passed to the automunge(.) function, are collectively saved and returned to the user in a dictionary that may serve as a key for consistent imputation basis to additional data in the postmunge(.) function, with such dictionary also serving as a key for any applied preprocessing transformations.

# 3 Preprocessing

The utility of the library extends well beyond missing data infill. Automunge is intended as a platform for all of the tabular learning steps following receipt of tidy data [6] (meaning one column per feature and one row per sample) and immediately preceding the application of machine learning. We found that by integrating the imputations directly into a preprocessing library, benefits included that imputations can be applied to returned multi-column categoric representations like one-hot encodings or binarized encodings, can account for potential data leakage between redundantly encoded feature sets, and can accept raw data as input as may include string encoded and date-time entries with only the minimal requirement of data received in a tidy form.

Under automation, Automunge normalizes numeric sets by z-score normalization and binarizes categoric sets (where binarize refers to a multi-column boolean integer representation where each categoric unique entry is represented by a distinct set of zero, one, or more simultaneous activations). We have a separate kind of binarization for categoric sets with two unique entries, which returns a single boolean integer encoded column (available as a single column by not having a distinct encoding set for missing data which is instead grouped with the most common entry). High cardinality categoric sets with unique entry count above a configurable heuristic threshold are instead applied with a hashing trick transform [7, 8], and for highest cardinality approaching all unique entries features are given a parsed hashing [9] which accesses distinct words found within entries. Further automated encodings are available for date-time sets in which entries are segregated by time scale and subject to separate sets of sine and cosine transforms at periodicity of time scale and additionally supplemented by binned activations for business hours, weekdays, and holidays. Designated label sets are treated a little differently, where numeric sets are left un-normalized and categoric sets are ordinal encoded (a single column of integer activations). All of the defaults under automation are custom configurable.

A user need not defer to automation. There is a built in extensive library of feature transformations to choose from. Numeric features may be assigned to any range of transformations, normalizations, and bin aggregations [10]. Sequential numeric features may be supplemented by proxies for derivatives [10]. Categoric features may be subject to encodings like ordinal, one-hot, binarization, hashing, or even parsed categoric encoding [11] with an increased information retention in comparison to one-hot encoding by a vectorization as a function of grammatical structure shared between entries. Categoric sets may be collectively aggregated into a single common binarization. Categoric labels may have label smoothing applied [12], or fitted smoothing where null values are fit to class distributions. Data augmentation transformations [10] may be applied which make use of noise injection, including several variants for both numeric and categoric features. Sets of transformations to be directed at a target feature can be assembled which include generations and branches of derivations by making use of our "family tree primitives" [13], as can be used to redundantly encode a feature set in multiple configurations of varying information content. Such transformation sets may be accessed from those predefined in an internal library for simple assignment or alternatively may be custom configured. Even the transformation functions themselves may be custom defined with only minimal requirements of simple data structures. Through application statistics of the features are recorded to facilitate detection of distribution drift. Inversion is available to recover the original form of data found preceding transformations, as may be useful to recover the original form of labels after inference.

Or of course if the data is received already numerically encoded the library can simply be applied as a tool for missing data infill.

## 4 Code Demonstration

Jupyter notebook install and imports are as follows:

```
!pip install Automunge

from Automunge import *
am = AutoMunge()
```

The automunge(.) function accepts as input a Pandas dataframe or tabular Numpy array of training data and optionally also corresponding test data. If any of the sets include a label column that header should be designated, similarly with any index header or list of headers to exclude from the ML infill basis. For Numpy, headers are the index integer and labels should be positioned as final column.

```
import pandas as pd
df_train = pd.read_csv('train.csv')
df_test = pd.read_csv('test.csv')
labels_column = '<labels_column_header>'
trainID_column = '<ID_column_header>'
```

These data sets can be passed to automunge(.) to automatically encode and impute. The function returns 10 sets (9 dataframes and 1 dictionary) which in some cases may be empty based on parameter settings, we suggest the following optional naming convention. The final set, the "postprocess_dict", is the key for consistently preparing additional data in postmunge(.). Note that if a validation set is desired it can be partitioned from df_train with valpercent and prepared on the train set basis. Shuffling is on by default for train data and off by default for test data, the associated parameter is shown for reference. Here we demonstrate with the assigncat parameter assigning the root category of a transformation set to some target column which will override the default transform under automation. We also demonstrate with the assigninfill parameter assigning an alternate infill convention to a column. The ML infill and NArw column aggregation are on by default, their associated activation parameters are shown for reference. Note that if the data is already numerically encoded and user just desires infill, they can pass parameter powertransform = 'infill'.

```
train, train_ID, labels, \
val, val_ID, val_labels, \
test, test_ID, test_labels, \
postprocess_dict = \
am.automunge(df_train,
             df_test = df_test,
             labels_column = labels_column,
             trainID_column = trainID_column,
             valpercent = 0.2,
             shuffletrain = True,
             assigncat = {'or23' : ['<parsed_categoric_target_column>'] },
             assigninfill = {'modeinfill' : ['<infill_target_column>'] },
             MLinfill = True,
             NArw_marker = True)
```

A list of columns returned from some particular input feature can be accessed with postprocess_dict['column_map']['<input_feature_header>']. A report classifying the returned column types (such as continuous, boolean, ordinal, onehot, binary, etc.) and their groupings can be accessed with postprocess_dict['columntype_report'].

If the returned train set is to be used for training a model that may go into production, the postprocess_dict should be saved externally, such as with the pickle library.

We can then prepare additional data on the train set basis with postmunge(.).

```
test, test_ID, test_labels, \
postreports_dict = \
am.postmunge(postprocess_dict,
             df_test)
```

## 5 Related Work

The R ecosystem has long enjoyed access to missing data imputation libraries that apply learned models to predict infill based on other features in a set, such as MissForest [1] and mice [14], where MissForest differs from mice as a deterministic imputation built on top of random forest and mice applies chained equations with pooled linear models and sampling from a conditional distribution. One of the limitations of these libraries are that the algorithms must be run through both training and inference for each separate data set, as may be required if test data is not available at time of training, which practice may not be amenable to production environments. Automunge on the other hand bases imputations on a designated train set, returning from application a collected dictionary of feature set specific models that can then be applied as a key for consistently preparing additional data on the train set basis.

Automunge's ML infill also differs from these R libraries by providing multiple auto ML options for imputation models. We are continuing to build out a range that currently includes Catboost [15], AutoGluon [16], and FLAML [17] libraries. Our default configuration is built on top of Scikit-Learn [5] random forest [4] models and may be individually tuned to each target feature with grid or random search by passing fit parameters to ML infill as lists or distributions.

There are of course several other variants of machine learning derived imputations that have been demonstrated elsewhere. Imputations from generative adversarial networks [18] may improve performance compared to ML infill (at a cost of complexity). Gaussian copula imputation [19] has a benefit of being able to estimate uncertainty of imputations. There are even imputation solutions built around causal graphical models [20]. Towards the other end of complexity spectrum, k-Nearest Neighbor imputation [21] for continuous data is available in common frameworks like Scikit-Learn.

Being built on top of the Pandas library, there is an inherent limitation that Automunge operations are capped at in-memory scale data sets. Other dataframe libraries like Spark [22] have the ability to operate on distributed datasets. We believe this is not a major limitation because the in memory scale is only associated with datasets passed to automunge(.) to serve as the basis for transformations and imputations. Once the basis has been established, transformations to any scale of data can be applied by passing partitions to the postmunge(.) function. We expect there may be potential to parallelize such an operation with a library like Dask [23] or Ray [24], such an implementation is currently intended as a future direction of research.

Another limitation associated with Pandas dataframes is that operations take place on the CPU. There are emerging dataframe platforms like Rapids [25] which are capable of GPU accelerated operations, which may particularly be of benefit when you take account for the elimination of a handoff step between main and GPU memory to implement training. Although the Pandas aspects of Automunge are CPU bound, the range of auto ML libraries incorporated are in some cases capable of GPU training for ML infill.

There will always be a simplicity advantage to deep learning libraries like Tensorflow [26] or PyTorch [27] which can integrate preprocessing as a layer directly into a model's architecture, eliminating the need to consider preprocessing in inference. We believe the single added inference step of passing data to the postmunge(.) function is an acceptable tradeoff because by keeping the preprocessing operations separate it facilitates a ML framework agnostic tabular preprocessing platform.

## 6 Experiments

Some experiments were performed to evaluate efficacy of a few different imputation methods in different scenarios of missing data. To amplify the impact of imputations, each of two data sets were pared down to a reduced set of the top 15 features based on an Automunge feature importance evaluation [11] by shuffle permutation [28]. (This step had the side benefit of reducing the training durations of experiments.) The top ranked importance categoric and numeric features were selected to separately serve as targets for injections of missing data, with such injections simulating scenarios of both missing at random and missing not at random.

To simulate cases of missing not at random, and also again to amplify the impact of imputation, the target features were evaluated to determine the most influential segments of the features' distributions

273 [29], which for the target categoric features was one of the activations and for the target numeric
274 features turned out to be the far right tail for both benchmark data sets.

275 Further variations were aggregated associated with either the ratio of full feature or ratio of distribution
276 segments injected with missing data, ranging from no injections to full replacement.

277 Finally, for each of these scenarios, variations were assembled associated with the type of infill
278 applied by Automunge, including scenarios for defaults (mean imputation for numeric or distinct
279 activations for categoric), imputation with mode, adjacent cell, and ML infill. The ML infill scenario
280 was applied making use of the CatBoost library to take advantage of GPU acceleration.

281 Having prepared the data in each of these scenarios with an automunge(.) call, the final step was to
282 train a downstream model to evaluate impact, again here with the CatBoost library. The performance
283 metric applied was root mean squared error for the regression applications. Each scenario was
284 repeated 68 or more times with the metrics averaged to de-noise the results.

285 Finally, the ML infill scenarios were repeated again with the addition of the NArw support columns
286 to supplement the target features.

# 7 Results

288 The results of the various scenarios are presented [Fig 2, 3, 4, 5]. Here the y axis are the performance
289 metrics and the x axis the ratio of entries with missing data injections, which were given as {0, 0.1,
290 0.33, 0.67, 1.0}, where in the 0.0 case no missing data was injected and with 1.0 the entire feature or
291 feature segment was injected. Because the 0.0 cases had equivalent entries between infill types, their
292 spread across the four infill scenarios are a good approximation for the noise inherent in the learning
293 algorithm. An additional source of noise for the other ratios was from the stochasticity of injections,
294 with a distinct set for each trial. Consistent with common sense, as the injection ratio was ramped up
295 the trend across infill scenarios was a degradation of the performance metric.

296 We did find that with increased repetitions incorporated the spread of the averaged performance
297 metrics were tightened, leading us to repeat the experiments at increased scale for some improved
298 statistical significance.

299 For the missing at random injections [Fig 2, 3], ML infill was at or near top performance across both
300 data sets, although the spread between imputations was not extremely pronounced. In most of the
301 setups, mode imputation and adjacent cell trended as reduced performance in comparison to ML infill
302 or the default imputations (mean for numeric sets and distinct activation set for categoric).

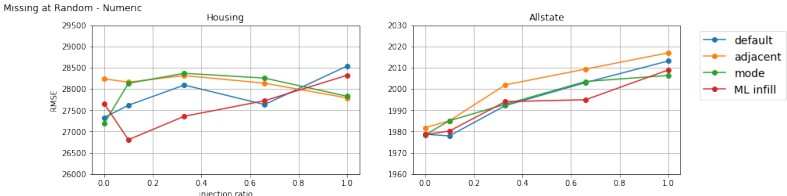

Figure 2: Missing at Random - Numeric Target Feature

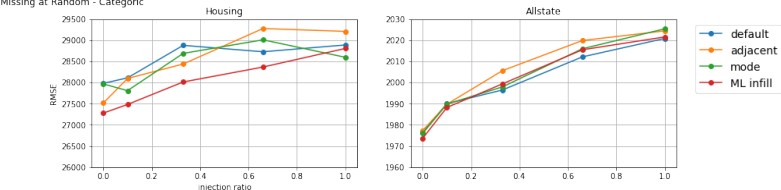

Figure 3: Missing at Random - Categoric Target Feature

303 For not at random injections to the right tail of numeric sets [Fig 4], it appears that ML infill had a
304 pronounced benefit to the Ames Housing data set [30], especially as the injection ratio increased,
305 and more of an intermediate performance to the Allstate Claims data set [31]. We speculate that ML
306 infill had some degree of variability across these demonstrations due to correlations (or lack thereof)

between the target feature and the other features, without which ML infill may struggle to establish a basis for inference. In the final scenario of not at random injections to categoric [Fig 5] we believe default performed well because it served as a direct replacement for the single missing activation.

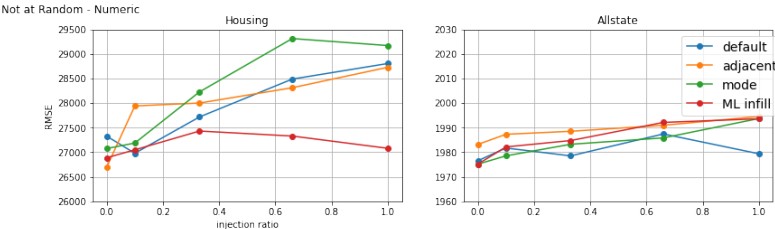

Figure 4: Not at Random - Numeric Target Feature

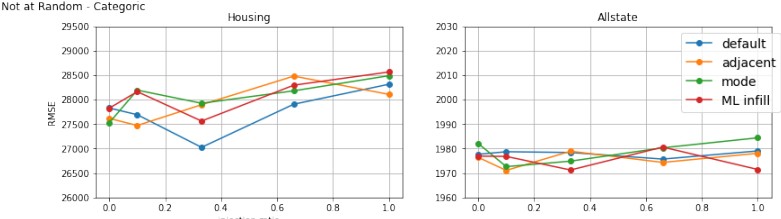

Figure 5: Not at Random - Categoric Target Feature

An additional comparable series of injections were conducted with ML infill and the added difference of appending the NArw support columns corresponding to the target columns for injections. Again these NArw support columns are the boolean integer markers for presence of infill in the corresponding entries which support the partitioning of sets for ML infill. The expectation was that by using these markers to signal to the training operation which of the entries were subjected to infill, there would be some benefit to downstream model performance. For many of the scenarios the visible impact was that supplementing with the NArw support column improved the ML infill performance, demonstrated here for missing at random [Fig 6, 7] and missing not at random [Fig 8, 9] with the other imputation scenarios shown again for context.

## 8    Discussion

One of the primary goals of this experiment was to validate the efficacy of ML infill as evidenced by improvements to downstream model performance. For the Ames Housing benchmark data set, there was a notable demonstration of ML infill benefiting model performance in the scenario of the numeric target column with not at random injections at increased injection ratios, and also to a lesser extent with missing at random injections. We speculate an explanation for this advantage towards the numeric target columns may partly be attributed to the fact that the downstream model was also a regression application, so that the other features selected for label correlation may by proxy have correlations with the target numeric feature. The corollary is that the more mundane performance of ML infill toward the categoric target columns may be a result of these having less correspondence with the surrounding features. The fact that even in these cases the ML infill still fell within noise distribution of the other imputation scenarios we believe presents a reasonable argument for defaulting to ML infill for tabular learning.

Note that as another argument for defaulting to ML infill as opposed to static imputations is that the imputation model may serve as a hedge against imperfections in subsequent data streams, particularly if one of the features experiences downtime in a streaming application for instance.

The other key finding of the experiment was the pronounced benefit to downstream model performance when including the NArw support column in the returned data set as a supplement to ML infill. This finding was consistent with our intuition, which was that increased information retention about infill points should help model performance. Note there is some small tradeoff, as the added training set dimensionality may increase training time. Another benefit to including NArw support columns may

be for interpretability in inspection of imputations. We recommend including the NArw support columns for model training based on these findings, with the one caveat that care should be taken to avoid inclusion in the data leakage scenario where there is some kind of correlation between presence of missing data and label set properties that won't be present in production.

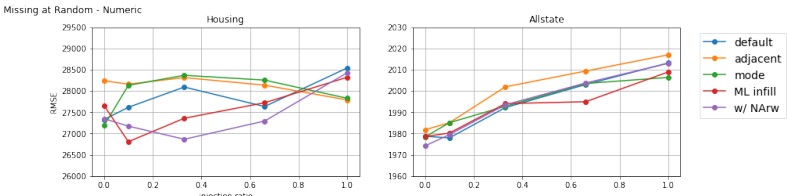

Figure 6: NArw comparison - Missing at Random - Numeric Target Feature

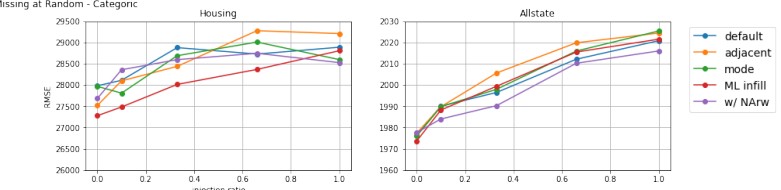

Figure 7: NArw comparison - Missing at Random - Categoric Target Feature

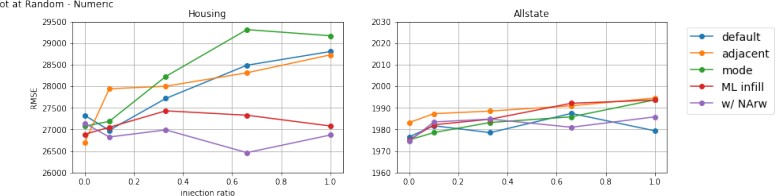

Figure 8: NArw comparison - Not at Random - Numeric Target Feature

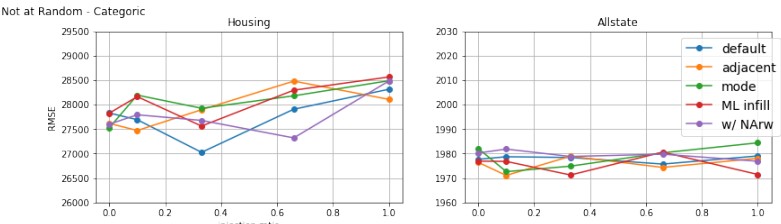

Figure 9: NArw comparison - Not at Random - Categoric Target Feature

# 9 Conclusion

Automunge offers a push-button solution to preparing tabular data for ML, with automated data cleaning operations like normalizations, binarizations, and auto ML derived missing data imputation aka ML infill. Transformations and imputations are fit to properties of a designated train set, and with application of automunge(.) a compact dictionary is returned recording transformation parameters and trained imputation models, which dictionary may then serve as a key for consistently preparing additional data on the train set basis with postmunge(.).

We hope that these experiments may serve as a kind of validation of defaulting to ML infill with supplemented NArw support columns in tabular learning for users of the Automunge library, as even if in our experiments the material benefits towards downstream model performance were not demonstrated for all target feature scenarios, in other cases there did not appear to be any material penalty. Note that ML infill can be activated for push-button operation by the automunge(.) parameter `MLinfill=True` and the NArw support columns included by parameter `NArw_marker=True`. Based on these findings these two parameter settings are now cast as defaults for the Automunge platform.

## Acknowledgments

A thank you owed to those facilitators behind Stack Overflow, Python, Numpy, Scipy Stats, PyPI, GitHub, Colaboratory, Anaconda, VSCode, and Jupyter. Special thanks to Scikit-Learn and Pandas.

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
