# OpenReview forum: "Missing Data Infill with Automunge"
_NeurIPS.cc/2021/Conference — NeurIPS 2021 Submitted_

### Official Review · Reviewer_Fgob · 2021-07-06

**Rating:** 2
**Confidence:** 5

**Summary:**

The paper introduces ‘ML infill’ , which is a Python implementation of a sequential imputation technique. More precisely, this method iteratively considers each feature $X_j$ with missing values as a target feature, fits a predictive model based on the remaining features as input variables and the observed values of $X_j$ as labels, and finally predicts the missing entries of $X_j$. ‘ML infill’ is provided as part of an open source Python library, Automunge, whose goal is to prepare data to feed it to a machine learning algorithm (e.g. feature transformation, and imputation of missing values). The paper provides experiments comparing the impact of basic imputations on downstream regression tasks, and recommend defaulting to ‘ML infill’ and using the missingness indicator as additional predictor.

**Limitations And Societal Impact:**

-

**Main Review:**

**Strengths**

The paper adresses an important problem for the application of machine learning in real-world settings: learning with messy tabular data. The proposed library is an effort to provide practitioners an automatic one-step preprocessing of messy data (notably when missing values are present) into tidy data that can be fed to off-the-shelves machine learning models.

**Weaknesses**

As noted by the authors in the Related work, sequential imputation methods that model each feature with missing values as a function of the others are not new (e.g. Mice, missForest, …).
More importantly, it seems to me that they are already available in Python thanks to Scikit-Learn’s IterativeImputer. In particular, similarly to the options provided by ‘ML infill’, IterativeImputer needs only be trained on the train set, can use a wide range of machine learning models for imputation, and also provides an option for stacking the missingness indicator with the imputed values.

Concerning the experiments, I think they are not extensive enough to provide recommendations with regards to the type of imputation method that’s best for downstream learning (only two datasets with 15 features selected). Also, it has already been observed by a number of works that in missing not at random settings, adding the missingness indicator as additional features can improve predictions (see for ex [1], [2], [3]).

For these reasons (availability of IterativeImputer in Scikit learn, weakness of experimental benchmark), I recommend rejection.
Also, it seems to me that since the imputation method is not novel, and that its implementation does not imply particular coding tricks, there is little novelty in the contribution.

Minor comment: l24-25, the missing data mechanism described is missing completely at random, which is different from missing at random.
[1] Sharafoddini et al A New Insight Into Missing Data in Intensive Care Unit Patient Profiles: Observational Study.
[2] Sperrin et al Missing data should be handled differently for prediction than for description or causal explanation.
[3] Josse et al On the consistency of supervised learning with missing values.

**Time Spent Reviewing:**

4

---

> ### Author Response · Authors · 2021-08-07
> **integration of ML imputations into a preprocessing platform**
>
> Hello reviewer, thank you for your feedback.
>
> In response to your summary:
> - Thank you for your summary, we believe this is well put. We would offer that perhaps it would be fair to also add that one of the contributions of this paper is that we are not only introducing ML infill, but that we are introducing the integration of ML imputations into a novel tabular preprocessing platform for applying univariate data transformations to the features in a tabular data set, as may be fit to properties of features in a training set for consistent basis on corresponding additional data, with data transformations as can be sourced from an extensive library or even custom defined with a simple template, and with transformations as may be applied under automation or with custom engineered data pipelines. We prepare tabular data for machine learning.
>
> In response to your Main Review - Strengths:
> - Please note that we believe an important omission in your description of strengths is the benefits associated with integration of ML imputations into a preprocessing platform, which we have noted to the other reviewers as including the ability to recognize features encoded in multicolumn representations and the ability to automatically account for potential data leakage between redundantly encoded feature sets. Our ability to apply imputations to received raw data based on an initial automated encoding was also novel when implemented, although it appears there is now a similar capability in a much reduced fashion with the DataWig imputation library that another reviewer noted.
>
> “an automatic one-step preprocessing of messy data (notably when missing values are present) into tidy data that can be fed to off-the-shelves machine learning models.”
> - As a clarification on this point, we assume data is received in a tidy form, which is shorthand for tabular data with one column per feature and one row per observation. However data returned from Automunge may not strictly adhere to the tidy data principle, as features may be redundantly encoded in multiple configurations (such as configurations with varying information content) and those configuration may include multicolumn representations (such as for example a categoric binarization).
>
> In response to your Main Review - Weaknesses:
>
> “they are already available in Python thanks to Scikit-Learn’s IterativeImputer”
> - As we noted to another review, our library predates IterativeImputer, and based on our review of their documentation IterativeImputer appears to be similar to k-means imputation in that it is suited for data sets with all continuous numeric features (as they treat each feature as a target for a regression model). Automunge allows imputations to mixed data sets that include both numeric and categoric features. Actually as quick aside we even support features received as datetime data types which we automatically encode with segregation of entries by time scale and separate sin and cos representations at periodicity of time scale.
>
> “Concerning the experiments, I think they are not extensive enough to provide recommendations with regards to the type of imputation method that’s best for downstream learning (only two datasets with 15 features selected).”
> - The purpose of the experiments was more to serve as a validation of our library rather than to draw broad conclusions. We hope the experiments could be considered successful from that standpoint. As we noted to another reviewer we applied the full resources at our disposal (a dedicated workstation) for several weeks to achieve the number of iterations needed to denoise the data.
>
> “it has already been observed by a number of works that in missing not at random settings, adding the missingness indicator as additional features can improve predictions (see for ex [1], [2], [3]).”
> - Thank you for drawing our attention to these works, sincerely. We propose that we could add citation to relevant works in a final print.
>
> “the missing data mechanism described is missing completely at random, which is different from missing at random.”
> - We agree that “missing completely at random” is a more accurate term than what we refer to as “missing at random”, we will make this correction in our final print.
>
>
> Please note that we have also provided additional detail in our responses to the other reviewers, and hope that you might take these into account for your final review. Please note that we are also currently working on additional improvements to our ML imputations methodology which we believe you will find of interest, and hope to have them rolled out prior to start of discussion period on 8/10. Best regards.

---

### Official Review · Reviewer_uuhM · 2021-07-13

**Rating:** 4
**Confidence:** 4

**Summary:**

This paper introduce a method called automunge to impute the missing entries in the data.
The imputation model is learned from the data and then the automunge imputes the missing values based on the learned model.

**Limitations And Societal Impact:**

This paper is too shallow that many important components and analysis were not explored. See my comments in the main review. I would recommend the authors to provide more details on the derivations, intuitions, and analysis on the automunge.

**Main Review:**

While I appreciate that the authors provided a nice summary of their method, there are many critical issues in missing data that were not answered.
1. [What is the underlying missing data assumption?]
A major issue is: an imputation model reflects our belief/assumption/model on the missing entries given the observed data. Different assumptions lead to different imputation distributions. This paper does not clearly indicate what is the underlying assumptions that they were making, so it is difficult to believe the imputed values.


2. [Deterministic vs Stochastic imputation]
When applying an imputation approach, we may impute the missing entries deterministically or stochastically. The later is generally preferred since deterministic imputation often leads to bias in the downstream analysis. See, e.g.,

Little, R. J., & Rubin, D. B. (2019). Statistical analysis with missing data (Vol. 793). John Wiley & Sons.

This should be discussed in the paper.


3. [Lack of details on the imputation method]
There is no details about which ML model to be used for imputation.
But this is a critical part of the analysis--different model reflects different assumptions on the missingness.


4. [Lack of analysis for the automunge]
The authors only describe how automunge works but there is no analysis on this approach. For a top ML conference, I don't think this is acceptable.



**Time Spent Reviewing:**

4

---

> ### Author Response · Authors · 2021-08-07
> **Imputations are now non-deterministic**
>
> Hello reviewer, thank you for your feedback.
>
> In response to your summary:
> “This paper introduce a method called automunge to impute the missing entries in the data.”
> As noted in comments to another reviewer
> - The full scope of Automunge was a bit too large to cover in detail in the paper, which was intended to offer an introduction with highlights to missing data infill aspects of the library. Automunge has attempted to consolidate the full range of the tabular learning workflow in between the two specific boundaries of 1) received “tidy data” (one column per feature and one row per sample) and 2) returned sets suitable for direct application of machine learning - by channeling through a single interface of a preprocessing platform built on top of the Pandas dataframe library. A helpful way to think of Automunge is that in addition to missing data infill it is a platform for applying univariate data transformations to tabular feature sets that may be fit to properties of a training set for consistent basis on additional data. Feature set transformations may be fairly simple, such as numeric normalization and categoric binarization under automation, or may be more elaborate.
> - Please note that data transformations are available to be assigned in sets that may include generations and branches of derivations, such as to present feature sets to ML training in multiple configurations of varying information content. Data transformations fit to properties of a train set may be applied under automation (such as based on an evaluation of data or distribution properties), may be sourced from an extensive internal library, or may even be custom defined by users (as we noted in sections 2 and 3) by use of a very simple template, which custom transformations may then be integrated into a pushbutton operation for preparing streams of data including transformations and imputations. We gave several examples of potential data transformations in section 3, including our own “parsed categoric encoding” (an alternative to one hot encoding) in which categoric features are vectorized based on grammatical structure shared between entries. We also noted several other benefits of channeling preprocessing through our interface, such as automated measurements for data distribution drift and an option for pushbutton data transformation inversions.
> - We noted in Section 3 that the integration of ML infill into a preprocessing platform enabled several benefits not available to other ML imputation libraries, such as the ability to recognize features encoded in multicolumn representations and the ability to automatically account for potential data leakage between redundantly encoded feature sets.
>
> In response to your Main Review:
>
> “1. What is the underlying missing data assumption?”
> - In our experiments we tried to simulate a wide range of missing data assumptions for validations in comparison to each. We simulated by engineering missing data injections scenarios of missing (completely) at random and missing not at random in targeted numeric and categoric features, each selected for their influence towards downstream model performance by way of an Automunge feature importance evaluation by shuffle permutation. Our goal was to validate the relative benefit of ML infill and NArw aggregation in comparison to each missing data assumption, such as to demonstrate that defaulting to these methods will often reach at or near top performance against the other imputation scenarios independent of assumptions, relying on the model training to navigate any peculiarities of a particular feature / missing data assumption.
>
> “2. Deterministic vs Stochastic imputation… we may impute the missing entries deterministically or stochastically. The later is generally preferred since deterministic imputation often leads to bias in the downstream analysis.”
> - Please note that we have recently rolled out some novel methods for incorporating stochasticity in our derived imputations, partly to serve as response to your feedback. Firstly we have introduced a default convention that learning algorithms that accept random seeds are fed a random sampled seed with each application, as opposed to prior convention that a uniform random seed was applied between each. This introduces some stochasticity into the model training aspects, although while still resulting in a deterministic imputation basis it at least encourages some diversity between iteration cycles. Secondly we have introduced a novel method for stochastic noise injections into derived imputations prior to insertion. The noise injections were partly inspired by some of our existing data transformations with noise injection we had available for uses like data augmentation and differential privacy (these noise injection transforms are discussed in the cited paper “Numeric Encoding Options with Automunge” with preprint included in supplemental material). We refer to this new method as “stochastic_impute”, which can be activated for noise injections to imputations associated with numeric or categoric features. Numeric noise injections sample from either a default normal distribution or optionally a laplace distribution. Default noise profile is mu=0, sigma=0.03, and flip_prob=0.06 (where flip_prob is ratio of a feature set's imputations receiving injections), each as can be custom configured. Please note that this noise scale is for injection to a min/max scaled representation of the imputations, which after injection are converted back to prior form, where the min/max scaling is based on properties of the feature in the training set. Please note that noise outliers are capped and noise distribution is scaled to ensure a consistent range of resulting imputation values is maintained in comparison to the feature properties in the training set.  Categoric noise injections sample from a uniform random draw from the set of unique activation sets in the training data (as may include one or more columns of activations for categoric representations), such that for a ratio of a feature's set's imputations based on the flip_prob (defaulting to 0.03 for categoric), each target imputation activation set is replaced with the randomly drawn activation set in cases of noise injection. We make use of numpy.random for distribution sampling in other cases. We propose that we could make note of this option in the paper and then provide full detail in a new appendix section. (Part of reason we preferred deterministic imputations prior was to support our software rollout validations which compare outputs of train and test data to ensure consistency, stochasticity kind of interferes with that.)
>
> “There is no details about which ML model to be used for imputation.”
> - To restate input provided to another reviewer: we’ve tried to offer different autoML library scenarios with different strengths which was not addressed at depth in the paper. Random forest was selected as the default due to simplicity, latency, and tendency not to overfit. We have seen benchmarks that suggests AutoGluon will likely outperform random forest, albeit with a high disk storage requirement associated with model ensembles and reduced latency performance. We included the FLAML library for their simplicity of setting a max training duration time through hyperparameter tuning, as well as their claimed performance from a latency standpoint. We selected CatBoost as a gradient boosting option and for GPU support. Partly we did not feel that we had sufficient rigor in our benchmarking to include discussions in the paper on this topic. We believe that an autoML library’s performance against generic tabular learning benchmarks should be a good proxy for their sufficiency to serve as an imputation model basis, and have tried to offer some diversity for user choice.
>
> “The authors only describe how automunge works but there is no analysis on this approach.”
> - We tried to balance a few agendas with this paper. One was to offer introduction and validation to ML infill with justification for casting as a new default in the library. Another agenda was to serve as an introduction to a fully featured preprocessing library that may serve as a replacement for instance to Scikit-learn for data pipelines, including demonstrations of code. We tried to clarify in the abstract that we considered one of the contributions of this paper to be the integration of ML imputation into a preprocessing platform, which platform has yet to be published in any formal venue.
>
> In response to your comments to Limitations And Societal Impact:
> “I would recommend the authors to provide more details on the derivations, intuitions, and analysis on the automunge.”
> - Please refer to any of the many preprints that we cited covering this territory, including for instance the paper “Parsed Categoric Encodings with Automunge” and “Numeric Encoding Options with Automunge”, each as available with the uploaded supplemental material. Outside of the preprints cited there is also a much more expansive backlog of writings that have covered similar material, however given their informal nature we decided not to include reference in this venue.
>
>
> Please note that we have also provided additional detail in our responses to the other reviewers, and hope that you might take these into account for your final review. Please note that we are also currently working on additional improvements to our ML imputations methodology which we believe you will find of interest, and hope to have them rolled out prior to start of discussion period on 8/10. Best regards.

---

### Official Review · Reviewer_mGQe · 2021-07-14

**Rating:** 2
**Confidence:** 4

**Summary:**

The paper presents the automunge library for missing data imputation and preprocessing for supervised learning in Python.
The automunge library implements ML-based infill similar to the established MissForest and MICE algorithms.

The authors discuss several imputation methods, as well as the automunge API, and evaluate the impact of different imputation strategies on two regression datasets, using a CatBoost model as the final machine learning model.

The authors conclude that ML infill with an indicator feature for missing values (called NArw in paper) achieves the best performance for continuous features, and that for categorical features, that all methods perform similarly.

**Limitations And Societal Impact:**

I don't see potential negative societal impact.

They authors did not address the limitations of their empirical evaluation, which I consider the main contribution of this work.

**Main Review:**

I think the authors address a really important problem. As far as I know, there is no extensive empirical evaluation of imputation methods for supervised learning, a gap that is crucial to close.
However, by restricting the evaluation to two regression datasets, I don't think it's possible for the authors to make very general claims. For the contribution to be empirical, I would expect at least a dozen datasets, ideally CC-18 or OpenML 100 for classification, and hopefully a similar set for regression.

Overall, the contributions of the work are not spelled out very clearly. There seem to be no significant methodological improvements over standard algorithms like MissForest. There is a discussion of the software library, however, the related work does not properly discuss related software work and how automunge fits into the ecosystem, so it seems not to be the focus of the contribution.
Finally, the empirical evaluation is very limited, as mentioned above.

## Detailed comments about software and related work
The core of automunge is ML-based imputation, which is actually implemented in scikit-learn as IterativeImputer.
This is not mentioned in the paper, even though it has been part of the library for a significant amount of time.
It's unclear what benefits automunge has over the implementation in scikit-learn. For the implementation or methodology to be a  contribution, this needs to be addressed.

Automunge seems to also allow automatic featurization, but there is very little detail about this in the paper, and there is no empirical evaluation.

It's unclear to me why automunge does not adhere to the standard scikit-learn interface for transformations (or estimators). In particular, given the interface as described in the paper, it's unclear to me how to use any of the standard cross-validation and grid-search utilities in scikit-learn.

## Notes on empirical evaluation
It's unclear how many iterations of ML infill have been applied for the benchmarks; I assume it's the default of 1.
I think one of the most critical questions is what model to use for infill, and what stopping criterion to use.
MissForest uses a somewhat odd heuristic for determining when to stop iterative infill. Scikit-learn uses the more natural heuristic of MICE, but this has been observed to never converge when using Random Forest base models.
Of course it is always possible to grid-search the number of iterations, but that is not an ideal outcome.

There are no error bars on the plots, which makes it hard to understand how much noise is in these observations. Given the ups and downs, I assume the plots to be quite noisy.

Finally, it's not clear why you would suggest ML infill for categorical variables if encoding it as a separate category behaves as well.


**Time Spent Reviewing:**

2

---

> ### Author Response · Authors · 2021-08-07
> **New early stopping criteria now implemented (1/2)**
>
> Hello reviewer, thank you for your feedback.
>
> In response to your summary:
> “The automunge library implements ML-based infill similar to the established MissForest and MICE algorithms.”
> - We note in Section 5 that one of the limitations of these particular imputation libraries is that “the algorithms must be run through both training and inference for each separate data set, as may be required if test data is not available at time of training, which practice may not be amenable to production environments.” Automunge fits imputation models on properties of a designated train set for consistent basis on additional data, with no need to retrain imputation models as more data becomes available.
>
> “The authors conclude that… for categorical features…”
> - Yes we’ve actually slightly tweaked the phrasing around categoric feature conclusions in our final preprint update after the NeurIPS deadline. A more accurate representation would be that we found the experiments demonstrated some benefit of ML infill towards missing at random categoric features and the results around missing not at random towards categoric target features did not demonstrate comparable benefit. (Reference figures 7 and 9 which include the ML infill both without (red) and with (purple) the NArw support column.) We noted that since even in cases where ML infill did not demonstrate pronounced benefit as performance still appeared to fall within noise distribution of the other imputation scenarios there did not appear to be any scenarios with material penalty associated with ML infill with NArw support column which was part of our justification of recommending as default for tabular learning.
>
> In response to your Main Review:
> “by restricting the evaluation to two regression datasets, I don't think it's possible for the authors to make very general claims”
> - Yes we agree with this point, we considered the main contribution of our paper the python library with ML infill imputation integrated into a preprocessing platform (see also our response to some of other reviewers). The experiments were primarily intended to serve as a validation as opposed to basis for general conclusions. We applied the full resources at our disposal for several months in running these experiments (including a few stops and restarts with refinement), which admittedly were not very extensive resources in the context of other researchers at this conference, and by resources am referring to the bulk of the training run on a dedicated workstation around the clock. The number of scenarios evaluated, coupled with the number of repetitions of each scenario to denoise, made this a lengthy process. We appreciate your suggestion of additional data sets (CC-18 or OpenML100) and hope that as additional resources become available may extend our investigations in some fashion.
>
> “There seem to be no significant methodological improvements over standard algorithms like MissForest.”
> - See our comments above in response to your summary as far as fitting imputation models to a train set for consistent basis on additional data. We also consider the integration of ML imputation into a preprocessing platform to be an important extension to other libraries by solving potential for data leakage in imputation model training under automation for redundantly encoded features.
>
> In response to your detailed comments about software and related work:
> “The core of automunge is ML-based imputation”
> - ML infill is an important component of the library, and was the main focus of this paper, but we believe a more accurate representation is that in addition to missing data infill the core of Automunge is a platform for applying univariate data transformations to tabular feature sets that may be fit to properties of a training set for consistent basis on additional data.
>
> “…implemented in scikit-learn as IterativeImputer. This is not mentioned in the paper, even though it has been part of the library for a significant amount of time. It's unclear what benefits automunge has over the implementation in scikit-learn.”
> - Based on your suggestion we reviewed the documentation for sklearn.impute.IterativeImputer. We suspect part of the reason we were unaware of this work was because our library predates this feature, which appears to have been rolled out in Scikit version 0.21. Based on our review, it appears their implementation is primarily intended for continuous numeric features as they only reference a regressor model, which would be consistent with other imputation options they offer like the k-Nearest Neighbor imputation we cited. Additionally there does not appear to be any form of preprocessing applied automatically, although they note that the imputer can be integrated into a Pipeline. They appear to have a more sophisticated stopping criterion for iterations of imputations than what is detail in the print you have reviewed, however Automunge has now rolled out our own early stopping criteria detail in this comment further below. So to be clear we believe Automunge has several material benefits over IterativeImputer, including support for both continuous and categoric features, integration into a preprocessing platform with automated encodings, and various other preprocessing aspects of the library that we noted in section 3.
>
> “Automunge seems to also allow automatic featurization, but there is very little detail about this in the paper, and there is no empirical evaluation.”
> - We cited several works that address Automunge featurization, with preprints provided in the supplemental material for anonymization purposes, including the papers “Parsed Categoric Encodings with Automunge” and “Numeric Encoding Options with Automunge”.
>
> 1/2

---

> > ### Author Response · Authors · 2021-08-07
> > **New early stopping criteria now implemented (2/2)**
> >
> > “It's unclear to me why automunge does not adhere to the standard scikit-learn interface for transformations (or estimators). In particular, given the interface as described in the paper, it's unclear to me how to use any of the standard cross-validation and grid-search utilities in scikit-learn.”
> > - We took the approach of building a preprocessing pipeline standard from the ground up (on top of the Pandas dataframe library), we have not sought to integrate into the Scikit-learn interface, please consider Automunge an alternative to Scikit for data transformation pipelines. We suggest applying our library as a precursor to any subsequent model training incorporating elements like cross-validation and grid-search. Please note though that we do actually have support for Scikit grid and random search in our imputation model training, currently available with the random forest default. We’ve developed an interface where any parameters passed to model training operations as lists / range / or distributions (as opposed to static values) automatically trigger a grid search or user may activate a random search for tuning the imputation model to each target feature. (We hope down the road to explore some more sophisticated hyperparameter tuning options, that is a future endeavor.)
> >
> > “It's unclear how many iterations of ML infill have been applied for the benchmarks”
> > - Yes confirmed we applied the default of 1 iteration.
> >
> > “I think one of the most critical questions is what model to use for infill”
> > - Yes agreed, we’ve tried to offer different autoML library scenarios with different strengths which was not addressed at depth in the paper. Random forest was selected as the default due to simplicity, latency, and tendency not to overfit. We have seen benchmarks that suggests AutoGluon will likely outperform random forest, albeit with a high disk storage requirement associated with model ensembles and reduced latency performance. We included the FLAML library for their simplicity of setting a max training duration time through hyperparameter tuning, as well as their claimed performance from a latency standpoint. We selected CatBoost as a gradient boosting option and for GPU support.  Partly we did not feel that we had sufficient rigor in our benchmarking to include discussions in the paper on this topic. We believe that an autoML library’s performance against generic tabular learning benchmarks should be a good proxy for their sufficiency to serve as an imputation model basis, and have tried to offer some diversity for user choice.
> >
> > “..and what stopping criterion to use.”
> > - We took your input as a clear signal that our library would benefit by some increased sophistication for stopping criterion in imputation iterations, which was prior based on specifying a hard coded integer of number of rounds. We thus have now rolled out our own version of early stopping for ML infill iterations. Our approach is based on comparing derived imputations of the current iteration to the preceding and deriving for all numeric features in aggregate and separately for all categoric features in aggregate a metric based on comparison of the imputations derived in the current iteration to the iteration preceding, with stopping conducted when both the numeric metric and categoric metric are within a configurable tolerance. Our numeric criteria has some similarity to approach in Scikit iterativeimputer, although we apply a different denominator in the formula (we believe the iterativeimputer formula may lose validity in scenario of presence of distribution outliers in their denominator set), and our categoric stopping criteria has some similarity to the MissForest approach, although we take a different approach of evaluating for metric within a tolerance instead of the sign of rate of change. More particularly, our numeric halting criteria is based on comparing for each numeric feature the ratio of max(abs(delta)) between imputation iterations to the mean(abs(entries)) of the current iteration, which are then weighted between features by the quantity of imputations associated with each feature and compared to a numeric tolerance value, and the categoric halting criteria is based on comparing the ratio of number of inequal imputations between iterations to the total number of imputations accross categoric features to a categoric tolerance value. Thank you for your input which helped us recognize the need for an early stopping criteria. We propose that we could add further detail on this implementation in the appendix and make note of it in the paper.
> >
> > “There are no error bars on the plots, which makes it hard to understand how much noise is in these observations.”
> > - Given the number of scenarios presented we thought error bars might result in too much clutter, we noted that since scenarios for 0% injection are comparable between imputation methods, their spread may serve as a proxy for noise inherent in the operation. Please note that since submitting this draft we have continued to perform additional experiment iterations in an attempt to further denoise the data, resulting in an increased minimum repetition count from 68 to 100. The results of these additional repetitions did result in a visible further “smoothing” of the performance curves, although I don’t believe NeurIPS allows me to share revisions as part of review process. (If we could go back we would eliminate the stochasticity of the validation split in the experiments, which was an unnecessary contributor of noise through repetitions.)
> >
> > “it's not clear why you would suggest ML infill for categorical variables if encoding it as a separate category behaves as well.”
> > - As you noted since these experiments were only performed on two data sets it's not possible to make very general claims, our experiment was intended as a validation, and ML infill did benefit categoric features in some of the scenarios (such as Figure 7). With further reflection on Figure 9, we consider the relatively flat performance curves with increasing injection ratio to be evidence that the narrow distribution segment associated with the not at random missing data injections did not have sufficient influence on downstream model performance for very a conclusive result. The not at random injections were applied by evaluating the range of unique entries of the feature and identifying the single most influential amongst them towards model performance by a shuffle permutation importance comparison of each unique entry in a one hot encoding. It appears this portion of experiment design may have benefited by selecting a target categoric feature set with fewer number of unique entries. (We had selected the feature for not at random injections to match the feature applied for random injections, which itself had been selected based on ranking categoric features by an Automunge feature importance evaluation by shuffle permutation, such that number of unique entries was not taken into account.)
> >
> > “…of their empirical evaluation, which I consider the main contribution of this work.”
> > - Please note that we consider the main contribution of this work to be a fully realized data preparation platform for tabular learning, which by integrating ML imputations into a preprocessing platform enables several benefits over competing libraries.
> >
> > Please note that we have also provided additional detail in our responses to the other reviewers, and hope that you might take these into account for your final review. Please note that we are also currently working on additional improvements to our ML imputations methodology which we believe you will find of interest, and hope to have them rolled out prior to start of discussion period on 8/10. Best regards.

---

> > > ### Comment · Reviewer_mGQe · 2021-08-21
> > > **Minor comments**
> > >
> > > > We suspect part of the reason we were unaware of this work was because our library predates this feature, which appears to have been rolled out in Scikit version 0.21.
> > >
> > > This has been released two years ago.
> > >
> > > >  Based on our review, it appears their implementation is primarily intended for continuous numeric features as they only reference a regressor model, which would be consistent with other imputation options they offer like the k-Nearest Neighbor imputation we cited.
> > >
> > > Indeed, that is the case, as there is no empirical evidence that imputing categorical variables is beneficial. For this to be a feature of AutoMunge, this would need to be demonstrated.
> > >
> > > > We suggest applying our library as a precursor to any subsequent model training incorporating elements like cross-validation and grid-search.
> > >
> > > This is not possible because imputation depends on the split of data into training and test set, so your library can not be used together with the cross-validation in scikit-learn and provide valid results. This is the main reason why I am skeptical of the current design.
> > >
> > > > Please note that we consider the main contribution of this work to be a fully realized data preparation platform for tabular learning, which by integrating ML imputations into a preprocessing platform enables several benefits over competing libraries.
> > >
> > > If this is the intention, then the paper would need to emphasize the data preparation platform and its features and how it compares with existing platforms like scikit-learn and the tools available in R, like recipes and MLR.

---

> > > > ### Author Response · Authors · 2021-08-21
> > > > **Response to your comments**
> > > >
> > > > Thank you for your additional comments. Please see our following direct responses:
> > > >
> > > > “This has been released two years ago.”
> > > >
> > > > - We have been publishing papers on Automunge for three years now on a regular basis, documenting nearly every major design decision, new feature, and experiment associated with the library. The ML infill integration dates back to the start of the project. We did not include reference to most of this material due to in most cases the form not adhering to the conventions of academic publishing. (Most of these writings followed the form of essays.)
> > > >
> > > > “there is no empirical evidence that imputing categorical variables is beneficial.”
> > > >
> > > > - We believe Figure 3 demonstrates a scenario where there is material benefit associated with ML infill as applied towards a categoric feature for missing completely at random.
> > > >
> > > > “imputation depends on the split of data into training and test set, so your library can not be used together with the cross-validation in scikit-learn and provide valid results.”
> > > >
> > > > - We don’t consider this a design flaw so much as an opportunity for further development. Please note that our current implementation already allowed for the simultaneous preparation of training and internally partitioned validation data, with validation data prepared separately on the basis of the training data (consistent with ‘test’ data).
> > > > - Based on your input we have now rolled out an extension to support external specification of specific partitions between train and validation data, which we believe was the only missing piece needed to support integration into a cross validation operation. (Previously our validation sets were either randomly sampled or accessed from bottom rows of the training data.)
> > > >
> > > > “the paper would need to emphasize the data preparation platform and its features and how it compares with existing platforms”
> > > >
> > > > - We devoted sections 3, 4, and 5 to covering similar topics. We were partly constrained by page count limits of this venue for a more extensive discussion.
> > > >
> > > > Best regards.

---

### Official Review · Reviewer_i1mF · 2021-07-16

**Rating:** 2
**Confidence:** 4

**Summary:**

The paper introduces a library for data wrangling. This include, in particular, feature process and missing value imputation. The library is described and demonstrated to be effective at the targeted tasks.

**Limitations And Societal Impact:**

Possible problems with missing value imputation are not addressed.

**Main Review:**

This is a well written paper that introduce a software library. It does not represent a novel scientific contribution. It also lacks comparison to competing methods, e.g. for missing value imputation.

## Some Detailed Comments
- In the abstract the authors "recommend" using the library they built. This seems to be a largely subjective judgement and is misplaced for a NeurIPS paper.
- The title is not great, does not give any information other than that the problem is about missing data imputation.
- l 24: The definition of "missing at random" seems wrong. What about "missing completely at random"?
- [DataWig](https://www.jmlr.org/papers/volume20/18-753/18-753.pdf) seems to be direct competition that is not considered/cited.

**Time Spent Reviewing:**

1

---

> ### Author Response · Authors · 2021-08-07
> **The full scope of Automunge**
>
> Hello reviewer, thank you for your feedback.
>
> In response to a few of your detailed comments:
>
> “In the abstract the authors "recommend" using the library they built. This seems to be a largely subjective judgement and is misplaced for a NeurIPS paper.”
> - One of the outcomes of this research as noted in the Conclusion section was the update of defaults in our preprocessing platform to activate ML infill unless otherwise specified, which was the reason for the declaration of our recommendation. We would offer that we could replace this recommendation statement in the abstract with a more concrete clarification that ML infill and NArw support column aggregation are now defaults in the library.
>
> “l 24: The definition of "missing at random" seems wrong. What about "missing completely at random"?”
> - We agree that “missing completely at random” is a more accurate term than what we refer to as “missing at random”, we will make this correction in our final print.
>
> “DataWig seems to be direct competition that is not considered/cited.”
> - Based on your suggestion we reviewed the paper “DataWig: Missing Value Imputation for Tables” by Felix Bießmann et al. We suspect part of the reason we were unaware of this work was because our library predates this paper. It appeared a unique aspect of their library was in their allowance for unstructured text entries, they reference the LSTM architecture and also converting to bag of words, the full details are not presented. (In Automunge we return unstructured text under automation by extracting distinct words and returning in a multi-column hashing.) We found a major limitation of the DataWig library was the very narrow range of data encodings, what they refer to as “featurizers”, which appear to be limited to numeric normalization, categoric one-hot encoding, and unstructured text to bag of words.
>
> Further clarification
> - The full scope of Automunge was a bit too large to cover in detail in the paper, which was intended to offer an introduction with highlights to missing data infill aspects of the library. Automunge has attempted to consolidate the full range of the tabular learning workflow in between the two specific boundaries of 1) received “tidy data” (one column per feature and one row per sample) and 2) returned sets suitable for direct application of machine learning - by channeling through a single interface of a preprocessing platform built on top of the Pandas dataframe library. A helpful way to think of Automunge is that in addition to missing data infill it is a platform for applying univariate data transformations to tabular feature sets that may be fit to properties of a training set for consistent basis on additional data. Feature set transformations may be fairly simple, such as numeric normalization and categoric binarization under automation, or may be more elaborate.
> - Please note that data transformations are available to be assigned in sets that may include generations and branches of derivations, such as to present feature sets to ML training in multiple configurations of varying information content. Data transformations fit to properties of a train set may be applied under automation (such as based on an evaluation of data or distribution properties), may be sourced from an extensive internal library, or may even be custom defined by users (as we noted in sections 2 and 3) by use of a very simple template, which custom transformations may then be integrated into a pushbutton operation for preparing streams of data including transformations and imputations. We gave several examples of potential data transformations in section 3, including our own “parsed categoric encoding” (an alternative to one hot encoding) in which categoric features are vectorized based on grammatical structure shared between entries. We also noted several other benefits of channeling preprocessing through our interface, such as automated measurements for data distribution drift and an option for pushbutton data transformation inversions.
> - We noted in Section 3 that the integration of ML infill into a preprocessing platform enabled several benefits not available to other ML imputation libraries, such as the ability to recognize features encoded in multicolumn representations and the ability to automatically account for potential data leakage between redundantly encoded feature sets.
>
> Regarding your notes on limitations and social impact:
> - Please note that we did discuss some aspects of limitations, particularly the caveat to avoid ML infill inclusion in the data leakage scenario where there is some kind of correlation between presence of missing data and label set properties that won’t be present in production.
>
> Please note that we have also provided additional detail in our responses to the other reviewers, and hope that you might take these into account for your final review. Please note that we are also currently working on additional improvements to our ML imputations methodology which we believe you will find of interest, and hope to have them rolled out prior to start of discussion period on 8/10. Best regards.

---

> > ### Comment · Reviewer_i1mF · 2021-08-24
> > **Thanks**
> >
> > Thanks for the detailed comments and clarification. While they are helpful, I don't think they touch upon the more general misfit of this work for NeurIPS.

---

> > > ### Author Response · Authors · 2021-08-25
> > > **A novel API and interface for working with data**
> > >
> > > Thank you reviewer. We recognize that the team behind this work is admittedly more accomplished at software development than writing academic papers. We hope you might consider the potential benefits to the machine learning community of our contributions of a completely novel API and interface for engineering tabular data pipelines through this open source platform.

---

### Author Response · Authors · 2021-08-09
**new automated data leakage detection**

This message intended for all reviewers:

Please note that as we were drafting our prior responses clarifying benefits of ML imputations as integrated into a preprocessing platform surrounding the capacity to automatically account for potential data leakage for redundantly encoded feature sets, it occurred to us that some further refinements could be implemented to automatically recognize other sources of data leakage for imputation models from the surrounding features. Particularly, it occurred to us that separate features with high prevalence of correlated missing data entries shared in common rows could be evidence that an imputation model including in basis that correlated feature could be a source of data leakage, since in imputation model training both features may have valid entries but in imputation model inference both features will have missing data.

Thus we have now implemented a method to compare aggregated NArw activations from a target feature in a train set to the surrounding features in a train set and for cases where a surrounding feature shares a high correlation of missing data based on the shown formula we exclude those surrounding features from the imputation model basis for the target feature.
((Narw1 + Narw2) == 2).sum() / NArw1.sum() > tolerance

We believe this automated detection for another source of data leakage could be considered another contribution of this library, and propose that we could make note of it in the paper and provide further detail in the appendix.

---

### Author Response · Authors · 2021-09-25
**additional validation findings**

Dear reviewers,

In an effort to validate repeatability of our findings, we have re-run the validation scenarios with ML infill targeting the categoric feature. We found a data quality issue impacting the second chart of Figures 5 and 9 that had resulted in slightly overstating the performance of ML infill for these scenarios. It is worth note that the first chart of Figures 3 and 7, which in the uploaded draft had shown a visible improvement for ML infill targeting missing at random injections to a categoric feature, did not fully replicate. There appears to have been some reversion to the mean and the new figures show ML infill for this scenario much closer to falling within noise distribution of the other infill scenarios, although still slightly beneficial. We propose using the revised figures and softening our language around this point. We believe there are still several meaningful contributions of this work outside of this validation, and hope that those may offset this update.

Regards

---

### Decision · Program_Chairs · 2021-09-27

**Decision:**

Reject

**Comment:**

The paper presents the automunge library for missing data imputation and preprocessing for supervised learning in Python. It has limited novelty. It tries to address the problem that currently there is no extensive empirical evaluation of imputation methods for supervised learning. However, it does not clearly indicate what are the underlying data assumptions, and lack details on the imputation methods. Empirical evaluation is performed on only two regression datasets. This is very limited and its conclusions are not convincing. It also lacks comparison to competing methods.